# Redirecting Imipramine against Bluetongue Virus Infection: Insights from a Genome-wide Haploid Screening Study

**DOI:** 10.3390/pathogens11050602

**Published:** 2022-05-22

**Authors:** Lijo John, Caroline Vernersson, Hyesoo Kwon, Ulrich Elling, Josef M. Penninger, Ali Mirazimi

**Affiliations:** 1National Veterinary Institute, 75189 Uppsala, Sweden; lijo.john@sva.se (L.J.); caroline.vernersson@sva.se (C.V.); hyesoo.kwon@sva.se (H.K.); 2IMBA—Institute of Molecular Biotechnology of the Austrian Academy of Sciences, 1030 Vienna, Austria; ulrich.elling@imba.oeaw.ac.at; 3Department of Medical Genetics, Life Science Institute, University of British Columbia, Vancouver, BC V6T 1Z3, Canada; josef.penninger@ubc.ca; 4Unit of Clinical Microbiology, Department of Laboratory Medicine, Karolinska Institute Karolinska University Hospital, 17177 Stockholm, Sweden

**Keywords:** bluetongue virus, haploid screening, imipramine, Niemann–Pick C1, BTV serotypes

## Abstract

Bluetongue virus (BTV), an arbovirus of ruminants, is a causative agent of numerous epidemics around the world. Due to the emergence of novel reassortant BTV strains and new outbreaks, there is an unmet need for efficacious antivirals. In this study, we used an improved haploid screening platform to identify the relevant host factors for BTV infection. Our screening tool identified and validated the host factor Niemann–Pick C1 (NPC1), a lysosomal membrane protein that is involved in lysosomal cholesterol transport, as a critical factor in BTV infection. This finding prompted us to investigate the possibility of testing imipramine, an antidepressant drug known to inhibit NPC1 function by interfering with intracellular cholesterol trafficking. In this study, we evaluated the sensitivity of BTV to imipramine using in vitro assays. Our results demonstrate that imipramine pretreatment inhibited in vitro replication and progeny release of BTV-4, BTV-8, and BTV-16. Collectively, our findings highlight the importance of NPC1 for BTV infection and recommend the reprofiling of imipramine as a potential antiviral drug against BTV.

## 1. Introduction

Bluetongue virus (BTV) is an economically important arboviral disease of domestic and wild ruminants that is transmitted by various species of midges [1,2]. Bluetongue disease can have severe impacts on the global livestock industry. The transmission of BTV in ruminants is mainly through the vector biting midges (*Culicoides* spp.) [3]. In general, the global distribution of BTV links closely to the favorable ecosystems of different midge species in temperate or tropical regions [4]. Currently, 36 distinct BTV serotypes have been identified in different ruminants [5,6,7,8]. Sheep are more vulnerable to BTV infection with high mortality rates, whereas cattle and goats are less affected. Cattle often remain asymptomatic and could serve as potential virus reservoirs of BTV [9]. However, BTV-8 is reported to cause severe disease and mortality in cattle [10]. Pathogenesis of BTV is like other viral hemorrhagic fevers, causing extensive endothelial dysfunction of vascular endothelium, resulting in hemorrhage, vascular occlusion, fluid exudations, and tissue infraction [11,12].

The emergence of novel reassorted BTV strains and the re-emergence of BTV strains make drug and vaccine development difficult [13,14,15]. Developing antivirals targeting the host factors might have greater potential against known and emerging serotypes of BTV. Some recent studies have explored the use of host-targeting drugs such as aurintricarboxylic acid [16] and fluvastatin against BTV in vitro [17,18]. On the other hand, direct-acting antivirals may be more challenging due to the mutation rates and emergence of new reassorting viruses. Therefore, these host factors for BTV must be identified to develop drugs blocking this host dependency for viral infection. To study the virus–host interface, haploid cells are powerful tools to uncover the biological process of various diseases [19]. Among the haploid screening tools, haploid mouse embryonic stem cells (mESCs) enable the possibility to perform both forward and reverse genetic screens. Indeed, haploid mESCs (haplobank) were previously used for identifying a major cellular factor involved in rhinovirus infection, PLA2G16. In this study, we utilized haploid mESCs to obtain an overview of cellular host factors required for BTV infection [20]. Our screening tool involves the isolation of virus-resistant cells, which lack the key host factors relevant for BTV infection. Here, we identify NPC1 as a major hit among the BTV-resistant haploid mESCs. NPC1 is an endo-lysosomal protein, involved in the cholesterol transport across lysosomes. Genetic dysfunctionality of NPC1 may cause fatal NPC disease, a lysosomal storage disorder, due to the accumulation of cholesterol in lysosomes [21]. We further show that NPC1 is a potential target for testing the existing drugs, and we found that imipramine, a tricyclic antidepressant, was able to inhibit BTV replication. Our findings demonstrate the utility of our screening tool in identifying NPC1 as a host factor for BTV and developing an antiviral drug against BTV.

## 2. Results

### 2.1. NPC1 Is a Host Factor for Bluetongue Virus Infection

To interrogate the novel hits for resistance against BTV infection, we performed a haploid screen with BTV-8 in mutagenized haploid mESCs (haplobank). BTV induces rapid cell death in infected haploid mESCs, which is beneficial for the selection of resistant cells. We infected 100 million mutagenized mESCs with BTV-8 and collected the surviving virus-resistant cells. Insertion sites in these BTV-resistant cells were mapped by sequencing and compared against the mutation frequencies in an unselected control cell population. All the sequencing analyses were carried out based on the number of independent insertions counted in the virus-resistant cells. Thus, the bubble size corresponds to the number of independent gene trap insertions. The small dots represent the hits with very low loss of function (LOF) score values. The y-axis is based on the significance −log *p* value. The highly scored gene hit in our screen was the NPC1 with the most insertions, which encodes an endo-lysosomal cholesterol transporter (Figure 1). The other hit, ST3GAL4, was not confirmed in our validation, and was therefore excluded from further analysis. To further confirm our findings, we extended our validation test in human diploid A549 cells lacking NPC1. The knockout status of NPC1 in different clones of A549 cells was confirmed by Western blot (Appendix A). We then infected the WT and NPC1 KO cells with BTV-8 and found that NPC1 KO cells were less susceptible to BTV-8 infection (Figure 2A, representative images in Appendix A). To determine whether NPC1 KO affects total virus yield, we checked viral titers in supernatants at 24 h post-infection with BTV-8. In NPC1 KO cells, there were significantly lower viral titers of BTV-8 (1.5 log) than in WT cells (Figure 2B). In addition to BTV-8, we also observed similar reductions in total viral titers of BTV-4 and BTV-16 in the supernatants (Appendix A). Overall, we identified and validated that NPC1 potentially plays a role in BTV infection.

### 2.2. Imipramine Inhibits BTV Infection

Structural and functional alterations in NPC1 are associated with several lysosomal storage disorders. The endo-lysosomal host–pathogen interface involving NPC1 is a critical host factor in the infection cycle of many viruses [22]. Based on our findings from haploid screening analysis, we tested whether the pharmacological intervention of NPC1 functions as a therapeutic target against BTV infection. For this study, we primarily focused on repurposing any of the FDA- and EMA-approved NPC1 inhibitors, since their safety profiles are known and more applicable. We chose imipramine, a tricyclic antidepressant drug in clinical use for humans and as an off-label drug in veterinary medicine. Before our in vitro analysis, we checked the toxicity of imipramine in A549 cells. Treatment of A549 cells with varying concentrations of imipramine ranging from 0 to 250 µM for 48 h showed no apparent cytotoxicity, with a 50% cytotoxic concentration (CC50) of 143 µM (Appendix A). To directly evaluate whether the BTV replication was affected, A549 cells were pretreated with imipramine for 24 h and infected with either BTV-4, BTV-8, or BTV-16. Impramine treatment was a combination of both pre- and post-infection treatment. Compared with DMSO-treated cells, the imipramine treatment resulted in a dose-dependent inhibition of intracellular BTV viral RNA levels by 2–3 log for BTV-4, BTV-8, and BTV-16 (Figure 3A). Table 1 summarizes the 50% cytotoxic concentration (CC50), 50% inhibitory concentration (IC50), and the selectivity index (SI) of imipramine in A549 cells. Infectious virus titers were examined in samples with the highest dose of imipramine (100 µM), which resulted in reduced virus yields (1.5–1.6 log10) of BTV-4, BTV-8, and BTV-16 (Figure 3B). We also tested the effect of 100 µM imipramine in type I IFN-deficient Vero cells [23], and the findings show that the antiviral activity of imipramine is not through the activation of IFN-related innate immunity (Figure 4). In addition, we examined the toxicity and antiviral effect of imipramine (100 µM) in ovine kidney cells to ensure that the antiviral effect is not restricted to human cells (Figure 4). Together, these results suggest that the effective concentration of imipramine has antiviral action against BTV.

## 3. Discussion

Bluetongue disease has severe economic impacts on the global livestock sector. BTV is widespread in tropical and subtropical regions, but mostly depends on the vector population in these regions. The currently available vaccines for BTV are serotype-specific, and thus do not offer cross-protection against multiple serotypes. The emergence of new serotypes and the lack of effective antivirals against all serotypes underscore the need for novel therapeutic strategies. Host-targeted antiviral drugs gain more attention because of their broader action against multiple serotypes and less chance of developing drug resistance. Most of these host-targeted therapies aim at host cellular pathways which are utilized by virus replication or through the innate immune against viruses. Moreover, these drugs can be developed in advance before the emergence of new virus strains.

Like other segmented viruses, BTV will undergo multiple gene assortments inside the multiple host serotypes, and this leads to the generation of new reassortant strains, making it difficult to predict during new outbreaks. Therefore, it is wise to develop host-targeted antivirals, and these will act as potential resources for future outbreaks. Along with unraveling new host–pathogen interactions of BTV, our goal was to identify potential targets to develop as antiviral therapies.

Unravelling the virus host biology of BTV infection may identify the potential targets for the development of host-targeted antiviral therapies. In our study, we used a haploid screening strategy by utilizing the haplobank constituting mouse embryonic stem cells to unravel the host–pathogen interactions of BTV. Previously, using this haplobank screen, PLA2G16 was identified as a pivotal host factor in rhinovirus infection. This advanced haploid library use the reversible technologies, in line with high throughput combining the power of stem cells with reproducible, functional annotation of the host genome. To our knowledge, BTV remains under-investigated to utilize the new information from the haploid screening studies for antiviral drug discovery. We used BTV-8 for our library screening with all the benefits of recapitulating the real infection process in BSL3 lab. Most significant bluetongue outbreaks in Europe are caused by BTV-8. Our screen identified NPC1 as a major factor for BTV-8. In addition, we also detected ST3 beta-galactoside alpha-2,3-sialyltransferase 4 (ST3GAL4), and this is involved in sialic acid biosynthesis. Sialic acid is a key receptor for BTV, and the outer capsid protein VP2 interacts with sialic acid during infection. Thus, our analysis underscores the power and unbiased way of our screening platform to identify the biologically significant candidate genes. Indeed, our screen did not identify a specific receptor, suggesting that BTV can use multiple routes for entry. Loss of NPC1 results in cholesterol and sphingolipid accumulation in lysosomes. In this study, we show that loss of NPC1 in human A549 cells exhibited moderate resistance to BTV-8, and affected the total virus yield in BTV-4 and BTV-16. Several lines of study have shown that NPC1 is a major entry factor for filoviruses, including Ebola and Marburg virus, while the release of HIV-1 and chikungunya is impaired in cells of patients with NPC disease. Functionally, NPC1 and NPC2 are the two major proteins that are involved in the intracellular transport of cholesterol in lysosomes. The accumulation of cholesterol in lysosomes leads to endosomal/lysosomal dysfunction and impairs the cholesterol metabolism, which is a hallmark of NPC disease. Based on these findings, our screening and validation results suggest the involvement of NPC1 during BTV infection.

Interestingly, NPC1 represents a promising host target for the development of antivirals against multiple viruses, and the possibility for repurposing the existing drugs that directly or indirectly target the NPC1 pathway. In this study, our goal was to redirect an FDA- or EMA-approved drug against BTV. Repurposing existing drugs will lower the development costs and shorten the time frames to use them to treat various disease conditions [24]. Therefore, we chose imipramine, a tricyclic antidepressant used for treating depression, also prescribed for off-label use in veterinary medicine [25,26,27,28,29]. In this study, we demonstrated that imipramine exhibits dose-dependent antiviral activity against BTV-4, BTV-8, and BTV-16. We could also validate the antiviral effect of imipramine against BTV in non-human cells such as Vero (monkey origin) and OK (ovine origin) cells. Imipramine is known to interfere in the intracellular cholesterol transport in endo-lysosomal compartments, so we speculate that this may hinder the virus release from the endo-lysosomal pathway. As a therapeutic intervention for BTV infection, imipramine or its analogues might be a potential drug against multiple serotypes of BTV. This could open doors to many new options for testing other analogues of imipramine or new combinations of drugs that target NPC1. Moreover, some imipramine analogues, e.g., clomipramine, are approved drugs for animal use [16].

Overall, our results support the repurposing of imipramine as a potential therapeutic drug against BTV. To date, there are no reports showing imipramine or its analogues as antiviral options against BTV infection. These drugs may be utilized in an outbreak situation to reduce the excess mortality in an epidemic. Our study requires further investigation by testing effective doses of imipramine or its analogues in animals. Most noteworthy, our genome-wide screening tool in this study will be instrumental for generating new potential leads for developing host-directed therapies against virus infections.

## 4. Materials and Methods

### 4.1. Cells and Viruses

Haploid mESCs used for this screening were derivatives of HMSc2, termed AN3-12, obtained from IMBA (Vienna, Austria) [20]. Haploid mESCs were cultured in standard ES cell medium, supplemented with 15% (*v*/*v*) fetal bovine serum (Hyclone), recombinant mouse Leukemia Inhibitory Factor (LIF R and D systems), and β-mercaptoethanol (Molecular Biology Grade—CAS 60-24-2—Calbiochem) (100 µM). A549 NPC1 knockout (KO) and wild-type (WT) cells were obtained from IMBA (Austria), and these cells were maintained in Dulbecco’s Modified Eagle Medium (D-MEM, Gibco) with 10% (*v*/*v*) fetal bovine serum (Hyclone), penicillin (100 μg/mL), and streptomycin (100 units/mL) (Gibco). Vero cells and Ovine kidney (OK) cells were maintained in EMEM supplemented with 10% FCS and L-glutamine containing penicillin–streptomycin.

In this study, we examined BTV-4, -8, and -16, which are among the most prevalent BTV serotypes in Europe. Vero cells were used to propagate BTV-4, -8, and -16 (passage number 2). All viruses (BTV-4, -8, and -16) used in our study were kindly provided by Stephan Zientara of the National Veterinary School in Alfort, France. All experiments with infectious viruses were performed at biosafety level 3 in compliance with biosafety guidelines (SVA, Uppsala).

### 4.2. Screen for BTV Host Factors Using Haploid mESCs

A haplobank library containing the haploid mESCs were used for the selection screens for genes required for BTV. Briefly, 100 million mESCs were seeded in 100 cm dishes and infected with BTV-8 at a multiplicity of infection (MOI) of 1 in 8mL of ES medium without FBS. After one hour, all plates were supplemented with complete ES medium incubated at 37 °C with 5% CO_2._ After 10–14 days, virus-resistant cells were isolated and expanded for 1 or 2 passages. These cells were subsequently pooled and used for genomic DNA isolation and for sequencing the gene trap sites. All sequencing analyses were conducted at IMBA (Vienna, Austria).

### 4.3. RT-qPCR

All RNA extractions were performed using the Direct-zol RNA miniprep kit (Zymo Research, Irvine, CA, USA), as described previously [30]. Quantitative real-time PCR reactions were carried out using a TaqMan Fast Virus 1-step Master Mix (Thermo Fisher, Waltham, MA, USA) and run on an Applied Biosystems 7500 machine. The following primer pairs were used in this study to detect BTV: BTV_IVI_F TGGAYAAAGCRATGTCAAA, BTV_IVI_R_ACRTCATCACGAAACGCTTC, and BTV_IVI_P probe ARGCTGCATTCGCATCGTACGC [31]. 18SrRNA (Thermo Fisher) was used as an endogenous control for normalization.

### 4.4. Immunofluorescence

A549 WT and NPC1 KO cells were infected with BTV at an MOI of 0.5. To visualize the infected cells, the cells were fixed in acetone for 30 min and incubated with primary antibody bluetongue virus (BTV) FITC conjugate MAb (CJ-F-BTV-MAB-10ML VMRD) (1/2000 dilution) at room temperature for 1 h. Cells were washed with PBS before incubation with DAPI (1/1000). Images of the infected cells were captured using a confocal laser scanning microscope (Zeiss LSM 800). To determine the number of BTV-infected cells, three representative wide-field 20× images were selected per experiment and at least 100 cells were counted. Infected cells were stained for BTV, and the percentage of infected cells was determined as the number of BTV-stained cells divided by the total number of cells ×100 [19].

### 4.5. Cytotoxicity Assay

A549, Vero, and OK cells were seeded in 96-well plates and treated with the indicated concentrations of imipramine hydrochloride. Then, 48 h post-treatment, cell viability was measured using the Cell Titer-Glo 2.0 assay (Promega, Madison, WI, USA). Viability is plotted as percentage of viability compared to the dimethyl sulfoxide (DMSO)-treated control. CC50 values of imipramine were calculated by the nonlinear regression analysis using GraphPad Prism 5.0 (GraphPad Software, San Diego, CA, USA).

### 4.6. Virus Titrations

Cell culture supernatants were collected from the BTV-infected wells 24 h post-infection. Tenfold serial dilutions of cell culture supernatants were prepared and added to the monolayer of Vero cells in a 96-well plate. After 24 h, viral titers were determined by immunofluorescence assay (IFA). The 50% tissue culture infectious dose (TCID_50_) was calculated based on the Spearman and Kärber algorithm, as described in [32].

### 4.7. Imipramine Treatment

Human A549 cells were pre-treated with either DMSO or increasing concentrations (25 µM, 50 µM, 75 µM, or 100 µM) of imipramine (Cat. No. HY-B1490 Medchemexpress) for 24 h. After 24 h, pre-treated cells were infected either with one of the following BTV serotypes (BTV-4, BTV-8, or BTV-16) at an MOI of 0.5 for 24 h in the presence of imipramine before further analysis.

### 4.8. Western Blot Analysis

Briefly, cells were lysed in lysis buffer, separated by SDS PAGE, and transferred onto polyvinylidene fluoride (PVDF) membranes (Thermo Fisher) using an iBlot 2 Gel Transfer Device. Membranes were blocked in 5% nonfat milk in 1X PBS, 0.1% Tween-20, and subsequently incubated overnight with primary antibody NPC1 (1:2000) at 4 °C. Antibodies against actin were used as loading control. After washing, membranes were incubated with secondary antibodies coupled to HRP and the signals were detected based on the enhanced chemiluminescence (ECL) method on an iBright (Thermo Fisher) imaging system.

## Figures and Tables

**Figure 1 pathogens-11-00602-f001:**
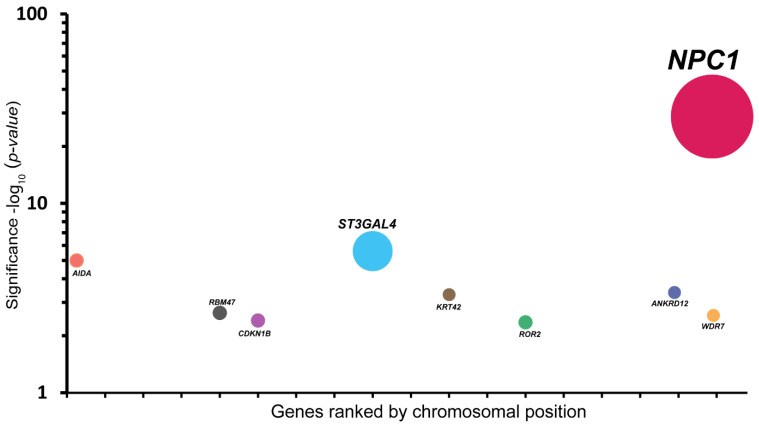
A haploid screen identifies BTV-resistant genes. Bubble plot of haploid screening screen analysis. Each bubble represents a gene, and the size corresponds to the number of gene trap insertions per gene. The x-axis shows the ranking of genes based on their chromosomal position. The y-axis shows −log of the *p*-value (0.001) of the total insertions in the gene compared to an unselected control.

**Figure 2 pathogens-11-00602-f002:**
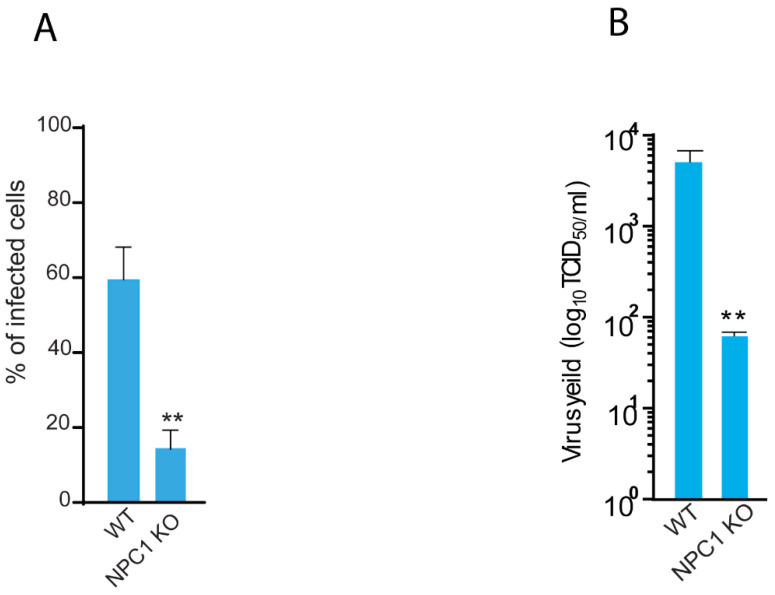
Loss of NPC1 impairs BTV infection and virus yield in human A549 cells. (**A**) WT control and NPC1 KO A549 cells were infected with BTV-8 (MOI 0.5) and the cells were fixed at 24 h post-infection. Infected cells were visualized using BTV monoclonal antibody conjugated with FITC. Results are represented as percentage of infected cells (*n* = 3, mean ± SD, Student’s *t*-test ** *p* < 0.001). (**B**) Total virus yield from the WT control cells and NPC1 KO cells infected with one of the BTV serotypes (BTV-4, BTV-8, and BTV-16) were determined by TCID_50_ titration on Vero cells (*n* = 3, mean ± SD, Student’s *t*-test ** *p* < 0.00).

**Figure 3 pathogens-11-00602-f003:**
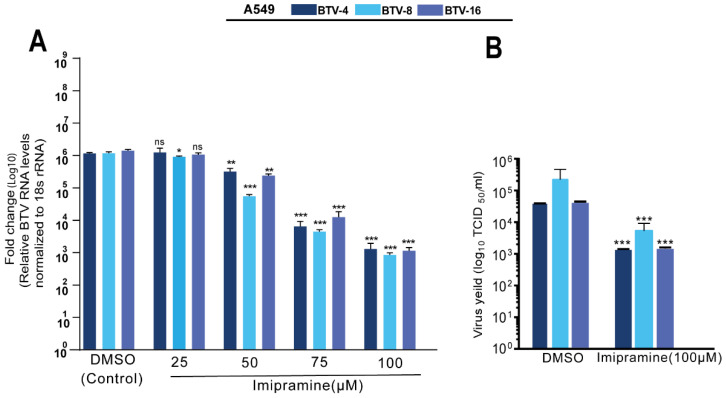
Imipramine treatment inhibits BTV replication in A549 cells. (**A**) A549 cells were pretreated either with varying concentrations of imipramine hydrochloride (25 to 100 µM) or with DMSO (0.4%) for 24 h and then challenged with either BTV-4, BTV-8, or BTV-16 (MOI 0.2). Inhibition of virus RNA was quantified using real-time RT-PCR, and the data are represented as fold change. 18srRNA was used as an internal control (*n* = 3, mean ± SD, Student’s *t*-test * *p* < 0.01, ** *p* < 0.001, *** *p* < 0.0001). (**B**) Inhibition of virus production in imipramine (100 µM) pretreated A549 cells was determined by 50% TCID_50_ titration in Vero cells. Results are the means of two independent experiments (*n* = 2, mean ± SD, Student’s *t*-test *** *p* < 0.0001).

**Figure 4 pathogens-11-00602-f004:**
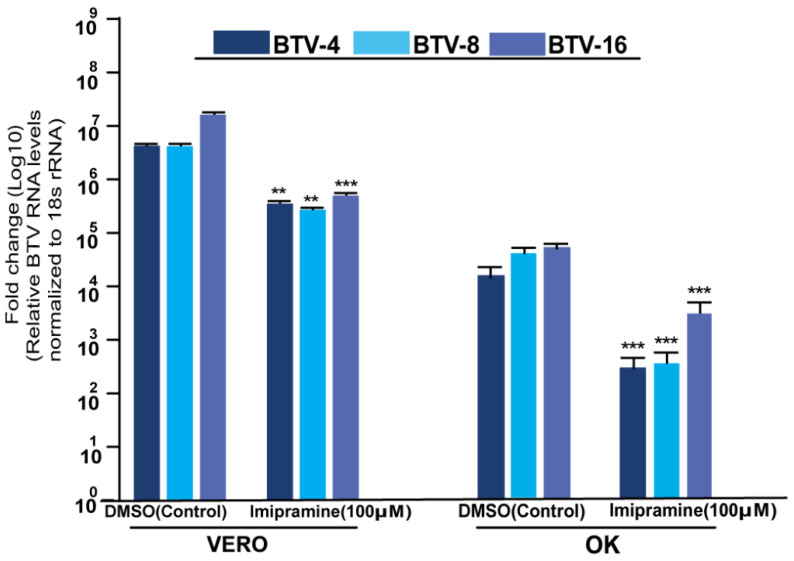
Imipramine treatment inhibits BTV infection in Vero and Ovine kidney (OK) cells. Vero and Ovine cells were pretreated with imipramine (100 µM) for 16–24 h and then challenged with either BTV-4, BTV-8, or BTV-16. RNA samples were extracted 24 h post-infection and BTV virus RNA was quantified using real-time RT-PCR, and the data are represented as fold change. 18srRNA was used as an internal control (*n* = 3, mean ± SD, Student’s *t*-test ** *p* < 0.001, *** *p* < 0.0001).

**Table 1 pathogens-11-00602-t001:** The CC50, IC50, and SI of imipramine.

Cells/Virus	CC50 (µM)	IC50 (µM)	SI (CC50/IC50)
BTV-4	-	45.51	3.1
BTV-8	-	30.56	4.6
BTV-16	-	33.01	4.2
A549	141.6	-	
Vero	128.4	-	
OK	160.7	-	

## Data Availability

The raw data generated in this study are available on request from the corresponding author.

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
