# Peer review of "Redirecting Imipramine against Bluetongue Virus Infection: Insights from a Genome-wide Haploid Screening Study"

_pathogens, 2022, doi:10.3390/pathogens11050602_

Round 1
Reviewer 1 Report
The manuscript entitled “Redirecting tricyclic antidepressant Imipramine against Blue- 2
tongue Virus infection: Insights from a genome wide haploid screening study” was reviewed.
This is an original study dealing with the assessment of imipramine as a candidate substance against BTV infection. More importantly, this substance targets host factors which is advantageous compared to direct antiviral drugs, the efficacy of which may be dependent on different viral serotype. The presentation of results including figures and figure legends is most appropriate.
The methodology followed assures the reliability of the results and it is adequately described.
The following minor issues need to be clarified and/or corrected.
The reason why the authors refer the EMA approval of imipramine in the abstract is unclear since this concerns a totally different use in different species. Is there any part of the research that have been done to support the registration of imipramine as an antidepressant drug that could have any connection with the present study?
In the discussion the authors stated that “Repurposing of existing drugs are faster,
cheaper, and reduced risks of failure in clinical development. Therefore, we choose imipramine, a tricyclic antidepressant used for treating depression, also prescribed for off
label use in veterinary medicine. This drug is mainly used in cats, dogs and horses to treat
multiple clinical conditions. In this study, we demonstrated that imipramine exhibits a
dose dependent antiviral activity against BTV 4, BTV 8 and BTV 16”. The authors are kindly asked to further explain this statement. What do they mean by clinical development? Does it refer to the pharmaceutical form and can it be the same for use in humans and ruminants? References are needed for the off-label use of the compound so as to enhance the validity of their claim.
The first paragraph of the introduction is not very well organized and there are quite a lot of repetition ie viral disease caused by virus. Moreover, the transmission is described twice although this is not so relevant with this study. Instead, it would have been much more relevant to add a brief description on the pathogenesis of the virus and a brief presentation of the clinical picture.
Author Response
Reviewer: The following minor issues need to be clarified and/or corrected.
The reason why the authors refer the EMA approval of imipramine in the abstract is unclear since this concerns a totally different use in different species. Is there any part of the research that have been done to support the registration of imipramine as an antidepressant drug that could have any connection with the present study?
Response: Thanks for these inputs to improve our manuscript. We have now rewritten major parts of the abstract and removed FDA/EMA. We have not done any registration of imipramine in connection with this study.
Reviewer: In the discussion the authors stated that “Repurposing of existing drugs are faster,cheaper, and reduced risks of failure in clinical development. Therefore, we choose imipramine, a tricyclic antidepressant used for treating depression, also prescribed for offlabel use in veterinary medicine. This drug is mainly used in cats, dogs and horses to treatmultiple clinical conditions. In this study, we demonstrated that imipramine exhibits adose dependent antiviral activity against BTV 4, BTV 8 and BTV 16”. The authors are kindly asked to further explain this statement. What do they mean by clinical development? Does it refer to the pharmaceutical form and can it be the same for use in humans and ruminants?
Response: Clinical development, is the total development time for a drug, and this may be shorter for the repurposed drugs compared to those associated with new chemical entities. The term drug repurposing/repositioning/reprofiling is frequently used in the literature eg:(Repurposing isoxazoline veterinary drugs for control of vector-borne human diseases PNAS www.pnas.org/cgi/doi/10.1073/pnas.1801338115 ), the human antidepressant mirtazapine is a repurposed form of mirataz. Most of the veterinary drugs come from repurposing the existing FDA/EMA approved human drugs, maybe with specifically formulated for animal use. Recent examples of human to animal translational success is the repurposed drug VDC-1101, for the treatment of canine lymphoma. We have chosen imipramine because it can inhibit NPC1 function, also this is used by veterinarians as an off-label drug (brand names: Tofranil®, Impril®).
Reviewer: References are needed for the off-label use of the compound so as to enhance the validity of their claim.
Response: Yes, we have added more references of imipramine use in horses and donkeys
- Turner, R. M.; Love, C. C.; McDonnell, S. M.; Sweeney, R. W.; Twitchell, E. D.; Habecker, P. L.; Reilly, L. K.; Pozor, M. A.; Kenney, R. M., Use of imipramine hydrochloride for treatment of urospermia in a stallion with a dysfunctional bladder. J Am Vet Med Assoc 1995, 207, (12), 1602-6.
Reviewer: The first paragraph of the introduction is not very well organized and there are quite a lot of repetition ie viral disease caused by virus. Moreover, the transmission is described twice although this is not so relevant with this study. Instead, it would have been much more relevant to add a brief description on the pathogenesis of the virus and a brief presentation of the clinical picture.
Response: We have added more about BTV pathogenesis in the introduction.
Reviewer 2 Report
The manuscript "Redirecting tricyclic antidepressant Imipramine against Blue-tongue Virus infection: Insights from a genome-wide haploid screening study" by Lijo John. Ref: pathogens-1705060 provides information on the pathogenetic aspects of the bluetongue virus (BTV), which are slight and/or unknown. Indeed, the authors identified and validated Niemman-pick C1 (NPC1). This protein is involved in lysosomal cholesterol transport as a critical factor in BTV infection, using an improved haploid screening platform to identify the relevant host factors for BTV infection. Furthermore, in the manuscript, a Food and Drug Administration (FDA)/European Medicines Agency (EMA)-approved antidepressant drug imipramine has been described as inhibiting in vitro of replication and progeny release of BTV-4, BTV-8, and BTV- 18.
After careful evaluation, I have some significant concerns regarding the publication of this article in the journal Pathogen.
Herein you can find the comments regarding the manuscript:
- According to the MDPI Instructions for Authors (https://www.mdpi.com/journal/pathogens/instructions#:~:text=Pathogens%20requires%20that%20authors%20publish,and%20references%20to%20unpublished%20data) the title of the manuscript should be concise, specific, and relevant. It should identify if the study reports (human or animal) trial data or is a systematic review, meta-analysis, or replication study. The current title is too long and dispersive. I suggest a more straightforward and more linear style for the title to focus the manuscript's topic on a few words.
- The Abstract is ambiguous. There is a certain distance between what is expressed in lines 12-16 and 16-21. Indeed, it is unclear why the authors tested the imipramine to inhibit in vitro replication of BTV-4, BTV-8, and BTV- 16.
- Section “Figure, Table and Schemes”: according to the MDPI Instructions for Authors (https://www.mdpi.com/journal/pathogens/instructions#:~:text=Pathogens%20requires%20that%20authors%20publish,and%20references%20to%20unpublished%20data), all Figures, Schemes and Tables should be inserted into the main text close to their first citation and must be numbered following their number of appearance (Figure 1, Scheme I, Figure 2, Scheme II, Table 1, ). Moreover, all Figures, Schemes, and Tables should have a short explanatory title and caption. The presence of the Figures, Schemes, and Tables in a reserved section makes it difficult to follow what is expressed by the authors.
- The section “Discussion” has the same issues highlighted in the "Abstract." I suggest structuring these sections better.
- In the "Materials and Methods," the authors should justify the choice to test only three serotypes (BTV-4, 8, and 16) of the 36 known serotypes, specifying the reasons that led to the choice of these specific serotypes and not others.
- I would encourage the authors to tone down the use of a new drug in animals herded to produce food for humans. Imipramine off-label use in cats, dogs, and horses, doesn’t justify the same benefit in ruminants. Furthermore, a molecule's in vitro antiviral activity does not mean maintaining the same antiviral activity in vivo.
Author Response
Herein you can find the comments regarding the manuscript:
- According to the MDPI Instructions for Authors (https://www.mdpi.com/journal/pathogens/instructions#:~:text=Pathogens%20requires%20that%20authors%20publish,and%20references%20to%20unpublished%20data) the title of the manuscript should be concise, specific, and relevant. It should identify if the study reports (human or animal) trial data or is a systematic review, meta-analysis, or replication study. The current title is too long and dispersive. I suggest a more straightforward and more linear style for the title to focus the manuscript's topic on a few words.
Response: Thanks for your suggestion. We have shortened the title as you suggested.
- The Abstract is ambiguous. There is a certain distance between what is expressed in lines 12-16 and 16-21. Indeed, it is unclear why the authors tested the imipramine to inhibit in vitro replication of BTV-4, BTV-8, and BTV- 16.
Response: We have rewritten the abstract and made necessary corrections in lines 19-23
- Section “Figure, Table and Schemes”: according to the MDPI Instructions for Authors (https://www.mdpi.com/journal/pathogens/instructions#:~:text=Pathogens%20requires%20that%20authors%20publish,and%20references%20to%20unpublished%20data), all Figures, Schemes and Tables should be inserted into the main text close to their first citation and must be numbered following their number of appearance (Figure 1, Scheme I, Figure 2, Scheme II, Table 1, ). Moreover, all Figures, Schemes, and Tables should have a short explanatory title and caption. The presence of the Figures, Schemes, and Tables in a reserved section makes it difficult to follow what is expressed by the authors.
Response: We have added short explanatory title and caption for the figures and now the figures are inserted into the main text close to their first citation.
- The section “Discussion” has the same issues highlighted in the "Abstract." I suggest structuring these sections better.
Response: Yes, we have modified the discussion part with the addition of more references.
- In the "Materials and Methods," the authors should justify the choice to test only three serotypes (BTV-4, 8, and 16) of the 36 known serotypes, specifying the reasons that led to the choice of these specific serotypes and not others.
Response: Yes, we have included those details in the materials and methods section.
- I would encourage the authors to tone down the use of a new drug in animals herded to produce food for humans. Imipramine off-label use in cats, dogs, and horses, doesn’t justify the same benefit in ruminants. Furthermore, a molecule's in vitro antiviral activity does not mean maintaining the same antiviral activity in vivo.
Response: Yes, we have discussed this issue in discussion part, also removed off-label use in cats, dogs in revised MS.
Round 2
Reviewer 2 Report
In the revised version of the manuscript, "Redirecting Imipramine against Bluetongue Virus infection: In-2 sights from a genome-wide haploid screening study " by Joh L. et al., the authors have satisfactorily addressed all my comments and suggestions. I have no other comments regarding the manuscript.